# ADAPTIVE CROSS-LAYER ATTENTION FOR IMAGE RESTORATION

## ABSTRACT

Non-local attention module has been proven to be crucial for image restoration. Conventional non-local attention processes features of each layer separately, so it risks missing correlation between features among different layers. To address this problem, we propose Cross-Layer Attention (CLA) module in this paper. Instead of finding correlated key pixels within the same layer, each query pixel is allowed to attend to key pixels at previous layers of the network. In order to mitigate the expensive computational cost of such hierarchical attention design, only a small fixed number of keys can be selected for each query from a previous layer. We further propose a variant of CLA termed Adaptive Cross-Layer Attention (ACLA). In ACLA, the number of keys to be aggregated for each query is dynamically selected. A neural architecture search method is used to find the insert positions of ACLA modules to render a compact neural network with compelling performance. Extensive experiments on image restoration tasks including single image super-resolution, image denoising, image demosaicing and image compression artifacts reduction validate the effectiveness and efficiency of ACLA.

## 1 INTRODUCTION

Image restoration algorithms aim to recover a high-quality image from a contaminated input image by solving an ill-posed image restoration problem. There are various image restoration tasks depending on the type of corruptions, such as image denoising (Zhang et al., 2019; Liu et al., 2018b), demosaicing (Zhang et al., 2017b; 2019), single image super-resolution (Fan et al., 2019; Lai et al., 2017; Tai et al., 2017), and image compression artifacts reduction(Zhang et al., 2017a). To restore corrupted information from the contaminated image, a variety of image priors (Buades et al., 2005; Zoran & Weiss, 2011; Zontak et al., 2013) were proposed.

Recently, image restoration methods based on deep neural networks have achieved great success. Inspired by the widely used non-local prior, most recent approaches based on neural networks (Zhang et al., 2019; Liu et al., 2018b) adapt non-local attention into their neural network to enhance the representation learning, following the non-local neural networks (Wang et al., 2018). In a non-local block, a response is calculated as a weighted sum over all pixel-wise features on the feature map to account for long-range information. Such a module was initially designed for high-level recognition tasks such as image classification, and it has been proven to be beneficial for low-level vision tasks (Zhang et al., 2019; Liu et al., 2018b).

Though attention modules have been shown to be effective in boosting the performance. Most attention modules only explore the correlation among features at the same layer. Actually, features at different intermediate layers encode variant information at different scales, and might be helpful to augment the information used in recovering the high-quality image. Motivated by the potential benefit of exploring feature correlation across intermediate layers, Holistic Attention Network (HAN) (Niu et al., 2020) is proposed to find interrelationship among features at hierarchical levels with a Layer Attention Module (LAM). However, LAM assigns a single importance weight to all features at the same layer and neglects the difference of spatial positions of these features. Recent research in omnidirectional representation (Tay et al., 2021) suggests that exploring the relationship among features at different layers can benefit the representation learning of neural networks. Nevertheless, calculating correlation among features at hierarchical layers is computationally expensive due to the quadratic complexity of dot product attention. The complexity of such cross-layer attention design is

increased from $(HW)^2 L$ to $(HWL)^2$, where $H, W$ are the height and width of the feature map and $L$ is the number of layers.

## 1.1 CONTRIBUTIONS

Our contributions are listed as follows.

First, in order to address the problem caused by only referring to keys within the same layer in most attention modules, we propose a novel attention module termed Cross-Layer Attention (CLA), which searches for keys across different layers for each query feature. With the help of the deformation mechanism, CLA only attends to a small set of keys at different layers for each query feature.

Second, we propose an improved CLA termed Adaptive Cross-Layer Attention (ACLA) which selects an adaptive number of keys at each layer for each query, and searches for the optimal insert positions of ACLA modules. We deploy ACLA modules on commonly used neural network models, e.g. EDSR (Lim et al., 2017a), for image restoration. Extensive experiments on single image super-resolution, image denoising, image compression artifacts reduction, and image demosaicing demonstrate the effectiveness of our approach.

## 2 RELATED WORKS

### 2.1 NEURAL NETWORKS FOR IMAGE RESTORATION

Adopting neural networks for image restoration has achieved great success by utilizing their power in representation. ARCNN (Dong et al., 2015a) was first proposed to use CNN for compression artifacts reduction. Later, DnCNN (Zhang et al., 2017a) uses residual learning and batch normalization to boost performance of CNN for image denoising. In IRCNN (Zhang et al., 2017b), a learned set of CNNs are used as denoising prior for other image restoration tasks. For single image super-resolution(Lai et al., 2017; Zhang et al., 2018d; Haris et al., 2018; Fan et al., 2019), even more efforts have been devoted to designing advanced architectures and learning methods. For example, RDN (Zhang et al., 2018d) and CARN (Ahn et al., 2018) fuse low-level and high-level features with dense connection to provide richer information and details for reconstructing. Recently, non-local attention (Liu et al., 2018b; Dai et al., 2019; Mei et al., 2020) is also used to further boost the peformance of CNN for image restoration.

### 2.2 ATTENTION MECHANISM

Attention mechanism has been applied to many computer vision tasks, such as image captioning (Xu et al., 2015; Chen et al., 2017) and image classification (Hu et al., 2018; Wang et al., 2017). Non-local attention (Wang et al., 2018) was first proposed to capture long-range dependencies for high-level recognition tasks. Recently, several works propose to leverage non-local attention for low-level vision tasks. In NLRN (Liu et al., 2018b) a recurrent neural network is proposed to incorporate non-local attention. RNAN (Zhang et al., 2019) proposed a residual local and non-local mask branch to obtain non-local mixed attention. RCAN (Zhang et al., 2018c) exploits the interdependencies among feature channels by generating different attention for each channel-wise feature. HAN (Niu et al., 2020) is proposed to find interrelationship among features at hierarchical levels with a layer attention module. Besides, some recent works attempt to explore the benefits of transformer based models for image restoration. IPT (Chen et al., 2021) is proposed to solve various restoration problems in a multi-task learning framework based on visual Transformer. SwinIR (Liang et al., 2021) adopt the architecture of Swin Transformer. However, compared with methods using CNN architecture, transformer-based image restoration methods usually use large datasets for training. Specifically, IPT uses ImageNet to pretrain the model. SwinIR adapts a combination of four datasets consisting of over 8000 high-quality images as training set for the tasks of denoising and compression artifact reduction.

### 2.3 NEURAL ARCHITECTURE SEARCH

Neural Architecture Search (NAS) has attracted lots of attention recently. Early works of NAS adopt heuristic methods such as reinforcement learning (Zoph & Le, 2016) and evolutionary algorithm (Xie & Yuille, 2017). The search process with such methods requires huge computational resources. Recently, various strategies are designed to reduce the expensive costs including weight sharing (Pham

et al., 2018), progressive search (Liu et al., 2018a) and one-shot search (Liu et al., 2018c; Xie et al., 2018). For example, DARTS (Liu et al., 2018c) firstly relaxes the search space to be continuous and conducts differentiable search. The architecture parameters and network weights are trained simultaneously by gradient descent to reduce the search time.

Despite the success of NAS methods for classification, dense prediction tasks such as semantic image segmentation and image restoration, usually demand more complicate network architectures. Some recent works have been devoted to explore hierarchical search space for dense prediction tasks. For example, Auto-DeepLab (Liu et al., 2019) introduces a hierarchical search space for semantic image segmentation. DCNAS (Zhang et al., 2021) build a densely connected search space to extract multi-level information. HNAS (Guo et al., 2020) also adopts a hierarchical search space for single image super-resolution.

## 3 METHOD

### 3.1 CLA: CROSS-LAYER ATTENTION

**Non-Local Attention.** Non-local attention (Wang et al., 2018) is designed to integrate self-attention mechanism into convolutional neural networks for computer vision tasks. It is usually applied on an input feature map $x \in \mathbb{R}^{H \times W \times C}$ to explore self similarities among all spatial positions of that feature map. For illustration purpose, we reshape $x$ to $N \times C$, where $N = H \times W$. A generic non-local attention can be formulated as

$$y_i = \frac{1}{C(x)} \sum_{n=1}^{N} f(x_i, x_n) g(x_n), \tag{1}$$

where $i$ indexes the spatial position of feature maps. $y$ is the output of non-local attention with the same size as $x$. $f(x_i, x_n)$ outputs the pairwise affinity between the query feature $x_i$ and its key feature $x_n$. $g(x_n)$ computes an embedding of feature $x_n$. $C(x)$ is a normalization term.

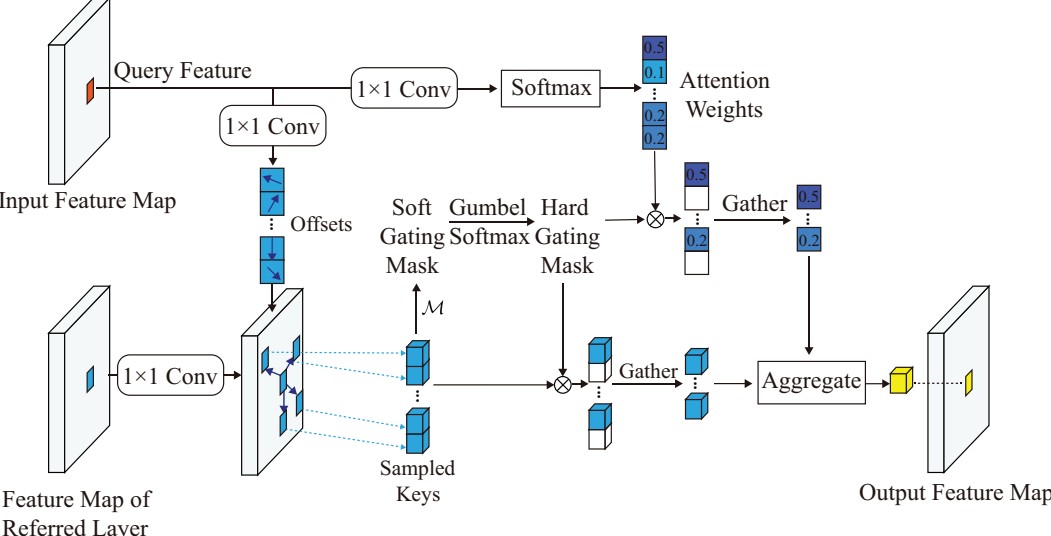

Figure 1: Illustration of Adaptive Cross-Layer Attention (ACLA) module. For each query pixel, a $1 \times 1$ convolution layer is used to obtain the offsets of the positions of keys sampled from the referred layer. Then, a convolution layer and Softmax are applied to the query feature to generate attention weights for sampled keys. To adaptively find the number of informative keys, a mask unit $\mathcal{M}$ together with the gumbel-softmax operation is used to generate a hard gating mask for the sampled keys. After the Hard Gating Mask is applied, the features of the selected keys are weighted by the corresponding attention weights. During inference time, a gather operation is applied on the selected keys.

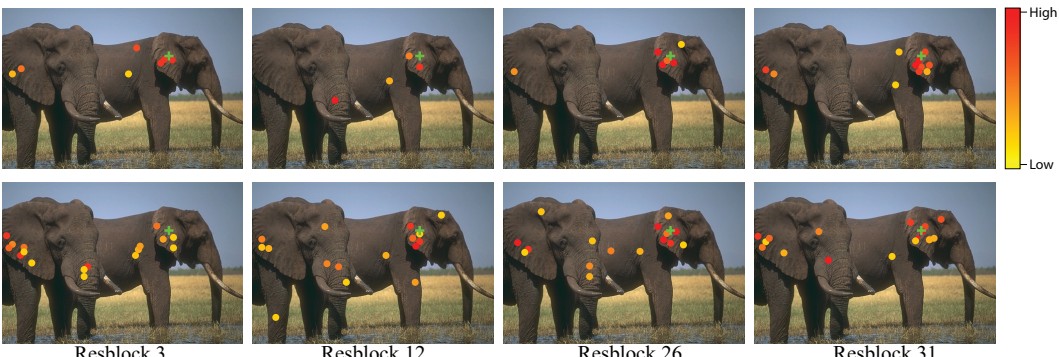

Resblock 3       Resblock 12       Resblock 26       Resblock 31

Figure 2: Visualization of selected keys by ACLA for a query feature from the 31st resblock. The first row shows the positions of the keys selected by ACLA with $K = 16$. For comparison, the positions of keys with top-16 attention weights following the vanilla Cross-Layer Non-Local attention formulation in Equation (3) is displayed in the second row. From left to the right are the sampled key positions from the 3rd, 12nd, 26th, and 31st resblock. The query feature is shown as green cross marker. Each sampled key feature is marked as a circle whose color indicates its attention weight. It can be observed that ACLA adaptively selects semantically similar key features for the query feature, and its vanilla counterpart lacks such capability. More visualization results can be found in Appendix A.5.

Non-local attention in Equation (1) is usually wrapped into a non-local block (Wang et al., 2018) with a residual connection from the input feature $x$. The mathematical formulation is given as

$$z = h(y) + x, \tag{2}$$

where $h$ denotes a learnable feature transformation.

**Cross-Layer Attention.** To search for keys from different layers for each query feature, we propose Cross-Layer Attention (CLA). To derive the formulation of CLA, we start by adapting non-local attention in Equation (1) to a cross-layer design. Suppose that $x^i$ is the output of the $i$-th layer in a CNN backbone for image restoration, where $i \in \{1, \ldots, L\}$ and $L$ is the number of layers. A non-local attention with cross-layer design can be formulated as

$$y_i^j = \frac{1}{\mathcal{C}(x^j)} \sum_{l=1}^{j} \sum_{n=1}^{N} f(x_i^j, x_n^l) g(x_n^l), \tag{3}$$

where the superscripts $j, l$ index the layer and the subscripts $i, n$ index the spatial locations of features. $y, x$ denote the output feature and input feature respectively. Query feature in this formulation can thus refer to features in previous layers. However, given the quadratic complexity of correlation computation, the complexity of such adaption of non-local attention is increased from $N^2 L$ to $(NL)^2$. In order to mitigate the expensive computation cost, we leverage deformable mechanism proposed in deformable convolution (DCN) (Dai et al., 2017) to design our proposed CLA as

$$y_i^j = \frac{1}{\mathcal{C}(x^j)} \sum_{l=1}^{j} \sum_{k=1}^{K} f(x_i^j, x^l(\boldsymbol{p}_i + \Delta \boldsymbol{p}_{ik})) g(x^l(\boldsymbol{p}_i + \Delta \boldsymbol{p}_{ik})), \tag{4}$$

where $k$ indexes the sampled keys. $\boldsymbol{p}_i$ represents the 2-d spatial position of the query in the feature map, and $\Delta \boldsymbol{p}_{ik}$ is the 2-d offset from the position of query $\boldsymbol{p}_i$ to the corresponding sampled key. As $\boldsymbol{p}_i + \Delta \boldsymbol{p}_{ik}$ can be fractional, bilinear interpolation is used as in (Dai et al., 2017) to compute $x(\boldsymbol{p}_i + \Delta \boldsymbol{p}_{ik})$. To further reduce the computation complexity in computing attention weights, attention weights can be generated from query feature alone. Thus, our proposed CLA can be simplified as

$$y_i^j = \frac{1}{\mathcal{C}(x^j)} \sum_{l=1}^{j} \sum_{k=1}^{K} f(x_i^j) g(x^l(\boldsymbol{p}_i + \Delta \boldsymbol{p}_{ik})), \tag{5}$$

where $f$ is the function to generate attention weights from the query feature. Similar to the design for non-local attention module in (Wang et al., 2018), when deploying CLA in CNN backbones, we also wrap it into a non-local block with residual connection.

With our proposed CLA, each query feature now can refer to a fixed number of keys from each previous layer. However, query features at different spatial positions may have different preferences on keys sampled from different layers. Especially for image restoration tasks, the restoration process at different spatial position may vary significantly due to the diversity of textures in an image. Besides, when deploying CLA in CNN backbones for image restoration, we find that increasing the number of inserted CLA will not constantly improve the performance, but result in much heavier computation cost. This observation motivates us to search for an optimal configuration of deploying CLA in neural networks to efficiently explore its representation learning capacity.

## 3.2 ACLA: Adaptive Cross-Layer Attention

To achieve adaptive key selection and the search for the optimal configuration of CLA, we propose Adaptive Cross-Layer Attention (ACLA). Specifically, for each query feature, we dynamically search for the keys sampled from previous layers with ACLA. Besides, when deploying ACLA in CNN backbones, a neural architecture search method is used to search for the insert positions of ACLA. An objective based on the computation cost of inserted ACLA modules is used to supervise the search procedure.

**Adaptive Cross-Layer Attention.** To adaptively search for the informative sampled keys for a query feature from its previous layers, we first follow the method in CLA to get a fixed number of sampled keys from each layer. Next, a hard gating mask is applied on each of the sampled key as

$$y_i^j = \frac{1}{\mathcal{C}(x^j)} \sum_{l=1}^{j} \sum_{k=1}^{K} m_{i,l}^k f(x_i^j) g(x^l(\boldsymbol{p}_i + \Delta\boldsymbol{p}_{ik})), \tag{6}$$

where $m_{i,l}^k$ is the hard gating mask for the $k$-th sampled key from $x^l$ for query feature $x_i^j$, whose value is either 1 or 0. We further relax the hard gating mask to continuous domain with the simplified binary Gumbel-Softmax (Verelst & Tuytelaars, 2020). Therefore, the hard gating mask $m_{i,l}^k$ is approximated by

$$\hat{m}_{i,l}^k = \sigma\Big(\frac{\beta_{i,l}^k + \epsilon_{i,l,1}^k - \epsilon_{i,l,2}^k}{\tau}\Big), \tag{7}$$

with sampling parameter $\beta_{i,l}^k$, Gumbel noise $\epsilon_{i,l,1}^k, \epsilon_{i,l,2}^k$, and temperature $\tau$. $\sigma$ is the sigmoid function. The sampling parameter $\beta_{i,l}^k$ can be regarded as a soft gating mask. To make the gating decisions on sampled keys input-dependent, a mask unit $\mathcal{M}$ is used to generate the soft gating mask $\beta$ from the features of the sampled keys as

$$\beta_{j,l}^k = \mathcal{M}(x^l(\boldsymbol{p}_i + \Delta\boldsymbol{p}_{ik})). \tag{8}$$

Following the design in (Verelst & Tuytelaars, 2020), a $1 \times 1$ convolution layer is used as the mask unit $\mathcal{M}$ in our model. To train the mask unit $\mathcal{M}$, the straight-through estimator from (Bengio et al., 2013; Verelst & Tuytelaars, 2020) is used for $m_{i,l}^k$. In particular, during backward pass, we set $m_{i,l}^k = \hat{m}_{i,l}^k$, while in forward pass we set

$$m_{i,l}^k = \begin{cases} 1 & \hat{m}_{i,l}^k > 0.5, \\ 0 & \hat{m}_{i,l}^k \leq 0.5. \end{cases} \tag{9}$$

Besides, Gumbel noise $\epsilon_{i,l,1}^k$ and $\epsilon_{i,l,2}^k$ are set to 0 during inference. To demonstrate the effectiveness of adaptive key selection in ACLA, we visualized keys selected by ACLA at different layers for a query feature in Figure 2

**Insert Positions.** To search for the insert positions of ACLA, we first densely insert ACLA after each layer of the CNN backbone as in Figure 3 to build the supernet. Similar to the gating formulation in ACLA, we define a hard decision parameter $s_j \in \{0, 1\}$ for the $j$-th inserted ACLA in the supernet. $s_j = 1$ indicates that a ACLA module is inserted after the $j$-th layer, and otherwise for $s_j = 0$. The simplified binary Gumbel-Softmax (Verelst & Tuytelaars, 2020) can also be used here to approximate the hard decision parameter $s_j$ by

$$s_j = \sigma\Big(\frac{\alpha_j + \epsilon_1^j - \epsilon_2^j}{\tau}\Big), \tag{10}$$

with sampling parameter $\alpha_j$, Gumbel noise $\epsilon$, and temperature $\tau$. Thus, the output of ACLA in the supernet can be expressed as

$$y_i^j = \frac{1}{\mathcal{C}(x^j)} \sum_{l=1}^{j} s_l \sum_{k=1}^{K} m_{i,l}^k f(x_i^j) g(x^l(\boldsymbol{p}_i + \Delta\boldsymbol{p}_{ik})). \tag{11}$$

Different from data-dependent design of the gating mask in ACLA, $\alpha_j$ here is defined as architecture parameter and can be directly optimized by stochastic gradient descent (SGD) during the search process. By gradually decrease the value of temperature, $\alpha_j$ will be optimized such that $s_j$ will approach 1 or 0.

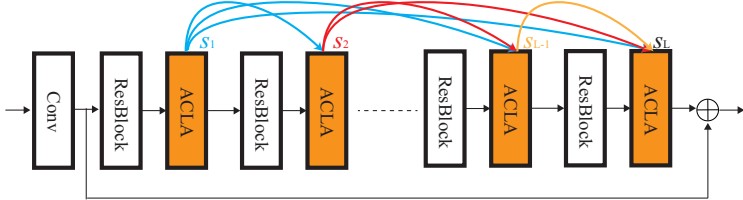

Figure 3: Illustration of the search for insert positions in ACLA

**Search Procedure.** We propose to optimize both the accuracy and the computation cost (FLOPs) of the ACLA modules inserted. Thus, the cost of the ACLA modules inserted needs to be estimated during the search phase. Following the formulation of the ACLA in the supernet, we can estimate the cost of the ACLA inserted after the $j$-th residual block as

$$\texttt{cost}_j = \sum_{l=1}^{j} s_l \sum_{k=1}^{K} (2m_{j,l}^k NC^2 + 2NC^2 + 6KNC), \tag{12}$$

where $N$ is the number of spatial positions. $C$ is the number of channels. $K$ is the maximal number of sampled keys. $2m_{j,l}^k NC^2$ is the FLOPs for the convolution on generating the gating masks. $2NC^2$ is the FLOPs for the convolution on sampling keys. $6KNC$ is the FLOPs for the aggregation process. Then we can get the estimation on the computation cost of all inserted ACLA modules as

$$\texttt{cost} = \sum_{j=1}^{L} s_j \texttt{cost}_j, \tag{13}$$

which is the summation of the cost of each layer, weighted by corresponding decision parameters. As mentioned before, with the help of the continuously relaxed representation in the supernet, we can search for the architecture by updating the architecture parameters using SGD. To supervise the search process, we design a loss function with the cost-based regularization to achieve the multi-objective optimization:

$$\mathcal{L}(w, \alpha) = \mathcal{L}_{MSE} + \lambda \log \texttt{cost}, \tag{14}$$

where $\lambda$ is the hyper-parameters that controls the magnitude of the cost term.

We find that at the beginning of the search process, ACLA modules inserted at shallow layers are more likely to be strengthened. To solve this problem, we split our search procedure into two stages. In the first stage, we only optimize the parameters of the network for enough epochs to get network weights sufficiently trained. In the second stage, we activate the architecture optimization. We alternatively optimize the network weights by descending $\nabla_w \mathcal{L}_{train}(w, \alpha)$ on the training set, and optimize the architecture parameters by descending $\nabla_\alpha \mathcal{L}_{val}(w, \alpha)$ on the validation set. When the search procedure terminates, we derive the insert positions based on the architecture parameters $\alpha$.

## 4 EXPERIMENTS

### 4.1 IMPLEMENTATION DETAILS

We select DIV2K (Timofte et al., 2017) as the training set for our experiments as like in (Dai et al., 2019; Zhang et al., 2018d), which include 800 images for training and 100 images for validation. We augment the training images by randomly rotating 90°, 180°, 270° and horizontally flipping. In each

Table 1: Quantitative results on benchmark datasets for single image super-resolution

| Method | Scale | Params(M) | Set5 | | Set14 | | B100 | | Urban100 | | Manga109 | |
|---|---|---|---|---|---|---|---|---|---|---|---|---|
| | | | PSNR | SSIM | PSNR | SSIM | PSNR | SSIM | PSNR | SSIM | PSNR | SSIM |
| Bicubic | ×2 | - | 33.66 | 0.9299 | 30.24 | 0.8688 | 29.56 | 0.8431 | 26.88 | 0.8403 | 30.80 | 0.9339 |
| SRCNN | ×2 | 0.244 | 36.66 | 0.9542 | 32.45 | 0.9067 | 31.36 | 0.8879 | 29.50 | 0.8946 | 35.60 | 0.9663 |
| VDSR | ×2 | 0.672 | 37.53 | 0.9590 | 33.05 | 0.9130 | 31.90 | 0.8960 | 30.77 | 0.9140 | 37.22 | 0.9750 |
| MemNet | ×2 | 0.677 | 37.78 | 0.9597 | 33.28 | 0.9142 | 32.08 | 0.8978 | 31.31 | 0.9195 | 37.72 | 0.9740 |
| SRMDNF | ×2 | 5.69 | 37.79 | 0.9601 | 33.32 | 0.9159 | 32.05 | 0.8985 | 31.33 | 0.9204 | 38.07 | 0.9761 |
| RDN | ×2 | 22.6 | 38.24 | 0.9614 | 34.01 | 0.9212 | 32.34 | 0.9017 | 32.89 | 0.9353 | 39.18 | 0.9780 |
| HAN | ×2 | 17.3 | 38.27 | 0.9614 | **34.16** | 0.9217 | 32.41 | 0.9027 | 33.35 | 0.9385 | 39.46 | 0.9787 |
| EDSR | ×2 | 40.7 | 38.11 | 0.9602 | 33.92 | 0.9195 | 32.32 | 0.9013 | 32.93 | 0.9351 | 39.10 | 0.9773 |
| EDSR+CLA | ×2 | 42.1 | 38.24 | 0.9613 | 34.08 | 0.9214 | 32.41 | 0.9028 | 33.28 | 0.9367 | 39.23 | 0.9780 |
| EDSR+ACLA | ×2 | 42.3 | **38.31** | **0.9617** | 34.10 | **0.9221** | 32.43 | **0.9030** | 33.35 | 0.9385 | 39.42 | 0.9787 |
| RCAN | ×2 | 15.3 | 38.27 | 0.9614 | 34.12 | 0.9216 | 32.41 | 0.9027 | 33.34 | 0.9384 | 39.44 | 0.9786 |
| RCAN+CLA | ×2 | 16.5 | 38.27 | 0.9615 | 34.14 | 0.9218 | 32.43 | **0.9030** | 33.34 | 0.9385 | 39.46 | 0.9785 |
| RCAN+ACLA | ×2 | 16.7 | 38.30 | 0.9615 | 34.15 | 0.9217 | **32.45** | 0.9029 | **33.39** | **0.9387** | **39.48** | **0.9789** |
| Bicubic | ×3 | - | 30.39 | 0.8682 | 27.55 | 0.7742 | 27.21 | 0.7385 | 24.46 | 0.7349 | 26.95 | 0.8556 |
| SRCNN | ×3 | 0.244 | 32.75 | 0.9090 | 29.30 | 0.8215 | 28.41 | 0.7863 | 26.24 | 0.7989 | 30.48 | 0.9117 |
| VDSR | ×3 | 0.672 | 33.67 | 0.9210 | 29.78 | 0.8320 | 28.83 | 0.7990 | 27.14 | 0.8290 | 32.01 | 0.9340 |
| MemNet | ×3 | 0.677 | 34.09 | 0.9248 | 30.00 | 0.8350 | 28.96 | 0.8001 | 27.56 | 0.8376 | 32.51 | 0.9369 |
| SRMDNF | ×3 | 5.69 | 34.12 | 0.9254 | 30.04 | 0.8382 | 28.97 | 0.8025 | 27.57 | 0.8398 | 33.00 | 0.9403 |
| RDN | ×3 | 22.6 | 34.71 | 0.9296 | 30.57 | 0.8468 | 29.26 | 0.8093 | 28.80 | 0.8653 | 34.13 | 0.9484 |
| HAN | ×3 | 17.3 | 34.75 | 0.9299 | 30.67 | 0.8483 | 29.32 | 0.8110 | 29.10 | 0.8705 | **34.48** | 0.9500 |
| EDSR | ×3 | 40.7 | 34.65 | 0.9280 | 30.52 | 0.8462 | 29.25 | 0.8093 | 28.80 | 0.8653 | 34.17 | 0.9476 |
| EDSR+CLA | ×3 | 42.1 | 34.75 | 0.9297 | 30.66 | 0.8481 | 29.30 | 0.8113 | 29.05 | 0.8700 | 34.33 | 0.9492 |
| EDSR+ACLA | ×3 | 42.3 | **34.76** | 0.9303 | 30.69 | 0.8484 | **29.34** | **0.8115** | 29.12 | 0.8706 | 34.40 | 0.9498 |
| RCAN | ×3 | 15.3 | 34.74 | 0.9299 | 30.65 | 0.8482 | 29.32 | 0.8111 | 29.09 | 0.8702 | 34.44 | 0.9499 |
| RCAN+CLA | ×3 | 16.5 | 34.75 | 0.9301 | 30.67 | **0.8485** | 29.31 | 0.8114 | 29.11 | 0.8705 | 34.46 | 0.9499 |
| RCAN+ACLA | ×3 | 16.7 | 34.74 | **0.9304** | 30.68 | **0.8485** | 29.33 | 0.8115 | **29.14** | **0.8709** | **34.48** | **0.9503** |
| Bicubic | ×4 | - | 28.42 | 0.8104 | 26.00 | 0.7027 | 25.96 | 0.6675 | 23.14 | 0.6577 | 24.89 | 0.7866 |
| SRCNN | ×4 | 0.244 | 30.48 | 0.8628 | 27.50 | 0.7513 | 26.90 | 0.7101 | 24.52 | 0.7221 | 27.58 | 0.8555 |
| VDSR | ×4 | 0.672 | 31.35 | 0.8830 | 28.02 | 0.7680 | 27.29 | 0.0726 | 25.18 | 0.7540 | 28.83 | 0.8870 |
| MemNet | ×4 | 0.677 | 31.74 | 0.8893 | 28.26 | 0.7723 | 27.40 | 0.7281 | 25.50 | 0.7630 | 29.42 | 0.8942 |
| SRMDNF | ×4 | 5.69 | 31.96 | 0.8925 | 28.35 | 0.7787 | 27.49 | 0.7337 | 25.68 | 0.7731 | 30.09 | 0.9024 |
| RDN | ×4 | 22.6 | 32.47 | 0.8990 | 28.81 | 0.7871 | 27.72 | 0.7419 | 26.61 | 0.8028 | 31.00 | 0.9151 |
| HAN | ×4 | 17.3 | 32.64 | 0.9002 | **28.90** | 0.7890 | 27.80 | 0.7442 | 26.85 | **0.8094** | **31.42** | 0.9177 |
| EDSR | ×4 | 40.7 | 32.46 | 0.8968 | 28.80 | 0.7876 | 27.71 | 0.7420 | 26.64 | 0.8033 | 31.02 | 0.9148 |
| EDSR+CLA | ×4 | 42.1 | 32.64 | 0.9001 | 28.83 | 0.7880 | 27.79 | 0.7435 | 26.79 | 0.8083 | 31.16 | 0.9159 |
| EDSR+ACLA | ×4 | 42.3 | 32.64 | **0.9003** | 28.88 | 0.7883 | **27.85** | 0.7443 | 26.87 | 0.8087 | 31.24 | 0.9174 |
| RCAN | ×4 | 15.3 | 32.63 | 0.9002 | 28.87 | 0.7889 | 27.77 | 0.7436 | 26.82 | 0.8087 | 31.22 | 0.9173 |
| RCAN+CLA | ×4 | 16.5 | **32.65** | 0.9002 | 28.88 | 0.7891 | 27.80 | 0.7441 | 26.86 | 0.8087 | 31.27 | 0.9177 |
| RCAN+ACLA | ×4 | 16.7 | 32.64 | 0.9002 | **28.90** | **0.7892** | 27.82 | **0.7445** | **26.89** | 0.8089 | **31.29** | **0.9179** |

mini-batch, 16 low-quality patches with size $48 \times 48$ are provided as inputs. ADAM optimizer is used for both the search phase and training phase. Default values of $\beta_1$ and $\beta_2$ are set as 0.9 and 0.999, respectively, and we set $\epsilon = 10^{-8}$. Before the search, we perform a cross-validation on 20% of the training data to decide the value of $\lambda$. In search phase, the learning rate is initialized as $10^{-4}$ and cosine learning rate schedule is used. The search process takes 200 epochs. In the training phase, the learning rate is initialized as $10^{-4}$ and then reduced to half every 200 epochs. The model is trained for 1000 epochs in total. $K = 8$ is sef for all CLA modules by default. For all ACLA modules, the value of $K$ are initialized as 16.

## 4.2 SINGLE IMAGE SUPER-RESOLUTION

We test our methods on top of the widely used super-resolution backbone EDSR (Lim et al., 2017b) and RCAN (Zhang et al., 2018b). The LR images are obtained by the bicubic downsampling of HR images. Our methods are evaluated on five standard datasets: Set5 (Bevilacqua et al., 2012), Set14 (Zeyde et al., 2010), B100 (Martin et al., 2001), Urban100 (Huang et al., 2015), and Manga109 (Matsui et al., 2017). The reconstructed results by our model are first converted to YCbCr space, PSNR and SSIM in the luminance channel is calculated in our experiments. We compare our methods with 6 other methods: SRCNN (Dong et al., 2015b), VDSR (Kim et al., 2016), MemNet (Tai et al., 2017), SRMDNF (Zhang et al., 2018a), RDN (Zhang et al., 2018d), and HAN (Niu et al., 2020). The quantitative results are shown in Table 1. Some visual results can be found in Appendix A.6. Our methods greatly improves the performance of EDSR and RCAN on all benchmarks and all upsampling scales. Besides, our methods outperform current state-of-the-art method HAN, which is also an attention based method, demonstrating the superiority of our proposed attention modules.

## 4.3 IMAGE DENOISING

We evaluate our proposed CLA and ACLA module on standard benchmarks for image denoising: KCLDAk24, BSD68 (Martin et al., 2001), and Urban100 (Huang et al., 2015). The noisy images

are created by adding AWGN noises with $\sigma = 10, 30, 50, 70$. We compare our approach with 4 methods, namely DnCNN (Zhang et al., 2017a), MemNet (Tai et al., 2017), RNAN (Zhang et al., 2019), and PANet (Mei et al., 2020). For fair comparisons, a 16-layer EDSR is used as the baseline CNN backbone. As shown in Table 2, our methods are also beneficial for representation learning with neural network on image denoising.

Table 2: Quantitative results on benchmark datasets for single image denoising

| Method | Params (M) | KCLDAk24 | | | | BSD68 | | | | Urban100 | | | |
|--------|-----------|------|------|------|------|------|------|------|------|------|------|------|------|
| | | 10 | 30 | 50 | 70 | 10 | 30 | 50 | 70 | 10 | 30 | 50 | 70 |
| MemNet | 0.677 | N/A | 29.67 | 27.65 | 26.40 | N/A | 28.39 | 26.33 | 25.08 | N/A | 28.93 | 26.53 | 24.93 |
| DnCNN | 0.672 | 36.98 | 31.39 | 29.16 | 27.64 | 36.31 | 30.40 | 28.01 | 26.56 | 36.21 | 30.28 | 28.16 | 26.17 |
| RNAN | 7.409 | 37.24 | 31.86 | 29.58 | 28.16 | 36.43 | 30.63 | 28.27 | 26.83 | 36.59 | 31.50 | 29.08 | 27.45 |
| PANet | 5.957 | 37.35 | 31.96 | 29.65 | 28.20 | 36.50 | 30.70 | 28.33 | 26.89 | 36.80 | 31.87 | 29.47 | 27.87 |
| baseline | 5.430 | 37.21 | 31.85 | 29.60 | 28.15 | 36.34 | 30.60 | 28.28 | 26.84 | 36.63 | 31.64 | 29.22 | 27.54 |
| CLA | 5.896 | 37.37 | **31.97** | 29.67 | 28.23 | 36.52 | 30.74 | 28.35 | 26.91 | 36.79 | 31.85 | 29.43 | 27.88 |
| ACLA | 5.914 | **37.38** | **31.97** | **29.70** | **28.25** | **36.54** | **30.77** | **28.36** | **26.94** | **36.85** | **31.90** | **29.49** | **27.91** |

## 4.4 IMAGE COMPRESSION ARTIFACTS REDUCTION

For the task of image compression artifacts reduction (CAR), we compare our method with 3 approaches: DnCNN (Zhang et al., 2017a), RNAN (Zhang et al., 2019), and PANet (Mei et al., 2020). All methods are evaluated on LIVE1 (Sheikh et al., 2005) and Classic5 (Foi et al., 2007). To obtain the low-quality compressed images, we follow the standard JPEG compression process and use Matlab JPEG encoder with quality $q = 10, 20, 30, 40$. For fair comparison, the results are only evaluated on Y channel in YCbCr Space. The quantitative results are shown in Table 3. 16-layer EDSR is used as our baseline CNN backbone. Both CLA and ACLA improves the performance of the neural network baseline.

Table 3: Quantitative results on benchmark datasets for image compression artifacts reduction

| Method | Params (M) | LIVE1 | | | | Classic5 | | | |
|--------|-----------|------|------|------|------|------|------|------|------|
| | | 10 | 20 | 30 | 40 | 10 | 20 | 30 | 40 |
| JPEG | - | 27.77 | 30.07 | 31.41 | 32.35 | 27.82 | 30.12 | 31.48 | 32.43 |
| DnCNN | 0.672 | 29.19 | 31.59 | 32.98 | 33.96 | 29.40 | 31.63 | 32.91 | 33.77 |
| RNAN | 7.409 | 29.63 | 32.03 | 33.45 | 34.47 | 29.96 | 32.11 | 33.38 | 34.27 |
| PANet | 5.957 | 29.69 | 32.10 | 33.55 | 34.55 | 30.03 | 32.36 | 33.53 | 34.38 |
| baseline | 5.430 | 29.63 | 32.04 | 33.50 | 35.51 | 29.99 | 32.22 | 33.43 | 34.31 |
| CLA | 5.896 | **29.73** | 32.13 | 33.57 | 35.54 | 30.05 | 32.38 | 33.55 | 34.42 |
| ACLA | 5.914 | **29.73** | **32.17** | **33.63** | **35.55** | **30.07** | **32.42** | **33.58** | **34.44** |

## 4.5 ABLATION STUDY AND DISCUSSION

**ACLA vs. Non-local attention** To verify the effectiveness of our proposed methods, we compare CLA and ACLA with Non-Local attention (Wang et al., 2018) and vanilla Cross-Layer Non-Local attention in terms of computational efficiency and performance. The vanilla Cross-Layer Non-Local attention follows the formulation in Equation (3). The comparison is performed on Set5 for single image super-resolution with EDSR backbone. The Non-Local attention modules and vanilla Cross-Layer Non-Local attention modules are inserted evenly after every 8th residual blocks. All the FLOPs in our ablation study are calculated for input of size $48 \times 48$. Results are presented in Table 4, where NL stands for Non-Local attention and CLNL stands for vanilla Cross-Layer Non-Local. As shown in Table 4, with less computation cost, CLA and ACLA achieve much better performance compared to standard Non-Local attention module.

Table 4: Efficiency and performance comparison with Non-Local attention on Set5

| Method | PSNR | FLOPs(G) | Params(M) |
|--------|------|----------|-----------|
| EDSR | 38.11 | 93.97 | 40.73 |
| NL | 38.15 | 109.38 | 43.56 |
| CLNL | 38.14 | 122.67 | 45.87 |
| CLA (Ours) | 38.24 | 96.93 | 42.13 |
| ACLA (Ours) | 38.31 | 96.97 | 42.29 |

**Number of inserted CLA modules** As stated before, the computation cost of the cross-layer design in CLA is quadratic to the number of inserted CLA modules. To verify that dense insertion of CLA

modules is not necessary, we perform an ablation study on the number of inserted CLA modules on Set5 ($\times$2) for single image super-resolution. We use EDSR as our backbone where $L$ CLA modules are evenly inserted. As shown in Table 5, with more CLA modules inserted, the performance can be slightly improved. However, the computation cost and parameter size of the model are also greatly increased. While with insert positions searched as shown in Table 6, our model can reach comparable performance with much less computational resources.

Table 5: Ablation study on number of inserted CLA modules on Set5

| Method | $L$ | PSNR | FLOPs(G) | Params(M) |
|--------|-----|------|----------|-----------|
| CLA | 4 | 38.24 | 96.93 | 42.13 |
| CLA | 8 | 38.27 | 101.37 | 44.35 |
| CLA | 16 | 38.27 | 118.48 | 51.47 |
| CLA | 32 | 38.26 | 182.93 | 79.29 |

**Ablation study on ACLA** As explained in Section 3.2, ACLA further improves CLA by two adaptive designs: selecting an adaptive number of keys at each layer for non-local attention and searching for optimal insert positions of ACLA modules. To verify the effectiveness of the two adaptive designs in ACLA, we perform an ablation study on their influence on top of CLA. The comparison is also performed on Set5 ($\times$2) for single image super-resolution with EDSR backbone. The results are shown in Table 6. CLA-I stands for CLA is deployed with the search for insert positions as in ACLA. CLA-K stands for CLA that adaptively select sampled keys as in ACLA. We can clearly see that both adaptive designs are beneficial to the performance of CLA.

Table 6: Ablation study on the effectiveness of insertion position search and sampled keys selection

| Method | Description of Methods | PSNR | FLOPs(G) | Params(M) |
|--------|------------------------|------|----------|-----------|
| CLA | - | 38.24 | 96.93 | 42.13 |
| CLA-I | search for insert positions | 38.27 | 96.93 | 42.13 |
| CLA-K | select aggregated keys | 38.28 | 96.87 | 42.29 |
| ACLA | - | 38.31 | 96.98 | 42.29 |

**Number of sampled keys $K$ for CLA and ACLA** As discussed before, the key point sampling strategy in CLA plays a vital role to reduce the computation cost. To verify that a small number of sampled keys $K$ can be sufficient, we perform experiments on CLA with different value of $K$. The comparison is performed on Set5 ($\times$2) for single image super-resolution with EDSR backbone. Besides, we also compare ACLA with different value $K$, which is the maximal number of sampled keys. The results are displayed in Table 7 and Table 8. With increased value of $K$, the performance of CLA and ACLA does not constantly improve. CLA with $K = 8$ and ACLA with $K = 16$ can already achieve comparable performance to those with larger $K$. This is also consistent with studies (Zhang et al., 2015; Elad & Aharon, 2006) on the power of sparse representation learning for image restoration.

Table 7: Ablation study on number of sampled keys in CLA on Set5

| Method | $K$ | PSNR | FLOPs(G) | Params(M) |
|--------|-----|------|----------|-----------|
| CLA | 4 | 38.22 | 96.71 | 42.09 |
| CLA | 8 | 38.24 | 96.93 | 42.13 |
| CLA | 16 | 38.25 | 97.38 | 42.21 |
| CLA | 32 | 38.23 | 97.90 | 42.39 |
| CLA | 64 | 38.25 | 98.92 | 42.74 |

Table 8: Ablation study on maximum number of sampled keys in ACLA on Set5

| Method | $K$ | PSNR | FLOPs(G) | Params(M) |
|--------|-----|------|----------|-----------|
| ACLA | 8 | 38.28 | 96.78 | 42.18 |
| ACLA | 16 | 38.31 | 96.98 | 42.29 |
| ACLA | 32 | 38.30 | 97.56 | 42.41 |
| ACLA | 64 | 38.31 | 98.03 | 42.69 |
| ACLA | 128 | 38.29 | 99.17 | 43.02 |

## 5 CONCLUSIONS

In this paper, we first propose a novel attention module Cross-Layer Attention (CLA) to search for informative keys across different layers for each query feature. We further propose Adaptive CLA, or ACLA, which improves CLA by two adaptive designs: selecting adaptive number of keys at each layer and searching for insert positions of ACLA modules. Experiments on image restoration tasks including single-image super resolution, image denoising, image compression artifacts reduction and image demosaicing validate the effectiveness and efficiency of CLA and ACLA.

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

# A APPENDIX

## A.1 MORE IMPLEMENTATION DETAILS

We summarize the value of $\lambda$, i.e. hyper-parameter that controls the magnitude of the cost term, for different experiments in Table 9. The insert positions of ACLA in the searched models are also displayed in the same table. Since RCAN constitutes 10 residual groups, we insert ACLA module after each of the residual group in the corresponding super network. For experiments with EDSR, ACLA modules are inserted after each residual block in the super network. Note that 16-layer EDSR is used for Image Denoising, Image Demosaicing, and Image Compression Artifacts Reduction.

Table 9: Search settings for ACLA in different experiments

| Task | Backbone | Value of $\lambda$ | Insert Positions |
|---|---|---|---|
| Single-Image Super-Resolution | EDSR | 0.15 | 3, 12, 26, 31, 32 |
| Single-Image Super-Resolution | RCAN | 0.3 | 1, 3, 5, 9 |
| Image Denoising | EDSR | 0.2 | 2, 7, 9, 13, 15 |
| Image Demosaicing | EDSR | 0.2 | 2, 5, 11, 14, 16 |
| Image Compression Artifacts Reduction | EDSR | 0.2 | 2, 7, 10, 13, 14 |

## A.2 IMAGE DEMOSAICING

To demonstrate the effectiveness of our proposed CLA and ACLA on various image restoration tasks, we add an experiment on image demosaicing. The evaluation is conducted on Kodak24, McMaster (Zhang et al., 2017b), BSD68, and Urban100, following the settings in RNAN (Zhang et al., 2019). We compare our approach with IRCNN (Zhang et al., 2017b), RNAN (Zhang et al., 2019), and PANet (Mei et al., 2020). We also use 16-layer EDSR as the baseline CNN model. As shown in Table 10, our approach yields the best reconstruction result for image demosaicing.

Table 10: Quantitative results on benchmark datasets for image demosaicing

| Method | Params(M) | McMaster18 | | Kodak24 | | BSD68 | | Urban100 | |
|---|---|---|---|---|---|---|---|---|---|
| | | PSNR | SSIM | PSNR | SSIM | PSNR | SSIM | PSNR | SSIM |
| Mosaiced | - | 9.17 | 0.1674 | 8.56 | 0.0682 | 8.43 | 0.0850 | 7.48 | 0.1195 |
| IRCNN | 0.731 | 37.47 | 0.9615 | 40.41 | 0.9807 | 39.96 | 0.9850 | 36.64 | 0.9743 |
| RNAN | 7.409 | 39.71 | 0.9725 | 43.09 | 0.9902 | 42.50 | 0.9929 | 39.75 | 0.9848 |
| PANet | 5.957 | 40.00 | 0.9737 | 43.29 | 0.9905 | 42.86 | 0.9933 | 40.50 | 0.9854 |
| Baseline | 5.430 | 39.81 | 0.9730 | 43.18 | 0.9903 | 42.66 | 0.9931 | 40.23 | 0.9852 |
| CLA | 5.896 | 40.03 | 0.9739 | 43.35 | 0.9906 | 42.88 | 0.9934 | 40.52 | 0.9853 |
| ACLA | 5.914 | **40.08** | **0.9742** | **43.38** | **0.9908** | **42.90** | **0.9936** | **40.55** | **0.9857** |

## A.3 COMPARISON WITH OTHER ATTENTION METHODS

In our paper, we have compared CLA and ACLA with non-local attention in section 4.5. Here, we further compare our proposed ACLA against other forms of attention modules that are widely used in the CV community, including Squeeze-and-Excitation (SE) (Hu et al., 2018) attention and Multi-Head Attention (MHA) (Bello et al., 2019). SE aim at reweighting the channel-wise responses by using soft self-attention to model interdependencies between the channels of the convolutional features. MHA is actually a variant of self-attention from the NLP domain. Specifically, MHA can be regarded as a special kind of non-local attention that takes account of the relative position information. For the experiments with SE and MHA, we insert four SE blocks and MHA blocks evenly in the EDSR backbone, respectively. The comparison is performed on 2x single-image super-resolution following the settings in section 4.2. The results are displayed in Table 12. Although MHA and SE bring improvements over the EDSR baseline. The best results are achieved by our proposed ACLA. Besides, we also perform an experiment that combines ACLA with SE, where SE blocks are inserted after each ACLA module.

Table 11: Efficiency and performance comparison with Squeeze-and-Excitation (SE) attention and Multi-Head Attention (MHA)

| Methods | Params(M) | FLOPs(G) | Set 5 | Set 14 | B 100 | Urban 100 | Manga 100 |
|---|---|---|---|---|---|---|---|
| EDSR | 40.73 | 93.97 | 38.11 | 33.92 | 32.32 | 32.93 | 39.10 |
| EDSR + MHA | 42.17 | 100.21 | 38.23 | 34.01 | 32.39 | 33.07 | 39.29 |
| EDSR + SE | 41.79 | 96.14 | 38.19 | 34.03 | 32.36 | 33.06 | 39.22 |
| EDSR + ACLA | 42.29 | 96.97 | 38.31 | 34.10 | 32.43 | 33.35 | 39.42 |
| EDSR + ACLA + SE | 43.47 | 99.32 | 38.33 | 34.09 | 32.44 | 33.38 | 39.46 |

## A.4 PERFORMANCE COMPARISON WITH OTHER ATTENTION-BASED METHODS

As image restoration is regarded as an ill-posed problem, improving the performance of CNN backbones for image restoration has always been a challenging task. Recently, attention methods have been widely used to improve performance. Here, we compare our ACLA with HAN and SAN, which are also attention-based methods for single image super-resolution. Both HAN and SAN are based on the previous state-of-the-arts (SOTA) method RCAN. In the table below, we compared the improvements of ACLA, HAN, and SAN. The improvements over RCAN are listed in the parentheses after the PSNR results. For our ALCA, the percentage comparisons of improvement are also calculated. For example, the improvement of ACLA over RCAN on B100 is 0.04, which is 400% of the improvement of SAN over RCAN. Besides, we have also calculated the improvement by ACLA on the EDSR backbone. As shown in the table, compared to HAN and SAN which are competitive baselines representing the recent progress of this literature, our method makes significant improvements in this literature.

Table 12: Efficiency and performance comparison with previous SOTA methods HAN and SAN for single-image super-resolution

| Methods | Inference Time | Set 5 | Set 14 | B100 | Urban100 | Manga109 |
|---|---|---|---|---|---|---|
| RCAN | 32.7 | 38.27 | 34.12 | 32.41 | 33.34 | 39.44 |
| HAN | 38.9 | 38.27 (0.00) | 34.16 (0.04) | 32.41(0.00) | 33.35 (0.01) | 39.46 (0.02) |
| SAN | 61.2 | 38.31 (0.04) | 34.07 (-0.05) | 32.42 (0.01) | 33.10 (-0.24) | 39.32 (-0.12) |
| RCAN+ACLA | 36.9 | 38.30 (0.03 / 75%) | 34.15 (0.03 / 75%) | 32.45 (0.04 / 400%) | 33.39 (0.05 / 500%) | 39.48 (0.04 / 200%) |
| EDSR | 16.2 | 38.11 | 33.92 | 32.32 | 32.93 | 39.10 |
| EDSR+ACLA | 19.8 | 38.31 (0.20) | 34.10 (0.18) | 32.43 (0.11) | 33.35 (0.42) | 39.42 (0.32) |

Besides, we also compare the inference time between our proposed ACLA, HAN, and SAN. The running time is the average of 1000 runs on input of size $48 \times 48$. The running time is evaluated on a single 16G Tesla V100. As shown Table 12, our method EDSR+ACLA achieves even better performance than HAN with much less inference time.

## A.5 VISUALIZATION OF SELECTED KEYS BY ACLA

We present more examples on visualization of selected keys by ACLA in Figure 4 to demonstrate the superiority of our method in searching for informative keys for query feature. The visualization is based on our results for $2\times$ image super-resolution. Similar to Figure 2, the first row shows the positions of the keys selected by ACLA with $K = 16$. For comparison, the positions of keys with top-16 attention weights following the vanilla CLNL attention formulation in Equation (3) is displayed in the second row. From left to the right are the sampled key positions from the 3rd, 12nd, 26th, and 31st resblock.

The visualization results show that ACLA adaptively selects semantically similar keys for the query feature, and its vanilla counterpart CLNL lacks such capability. For instance, in Figure 2, the query is from the ear of the elephant on the right side. With ACLA, 60% of the selected keys across are also from the ear of the same elephant. Besides, among the keys selected outside the ear of the same elephant, 5 out of 11 are from the ear of the elephant on the left, which have similar textures as the ear of the elephant on the right. While with CLNL, only 39% of the selected keys are from the the ear of the elephant on the right. Similar observations can also be found in Figure 4. In Figure 4 (a), we pick a query point from the frame structure at the top of a gate. With ACLA, 90% of the keys selected are distributed on the frame structures at the top of gates. While with CLNL, positions from the gates and the frame structure at the balcony are also given high attention weights. Only 61% of the selected keys are distributed on the frame structures at the top of gates, which may limit the power of attention

modules. Similar observations can also be found Figure 4 (b) and Figure 4 (c). In Figure 4 (b), the query is from the bridge in the middle of the image. All the keys selected by ACLA are also from the bridge. In Figure 4 (c), the query is from the back of a yak. Most of the keys selected by ACLA are also located on the body of yaks. While as shown in the second row, CLNL even assigns large attention weights to positions from the grass and the background. Such observations strongly demonstrate the power of ACLA in searching for informative keys across different layers.

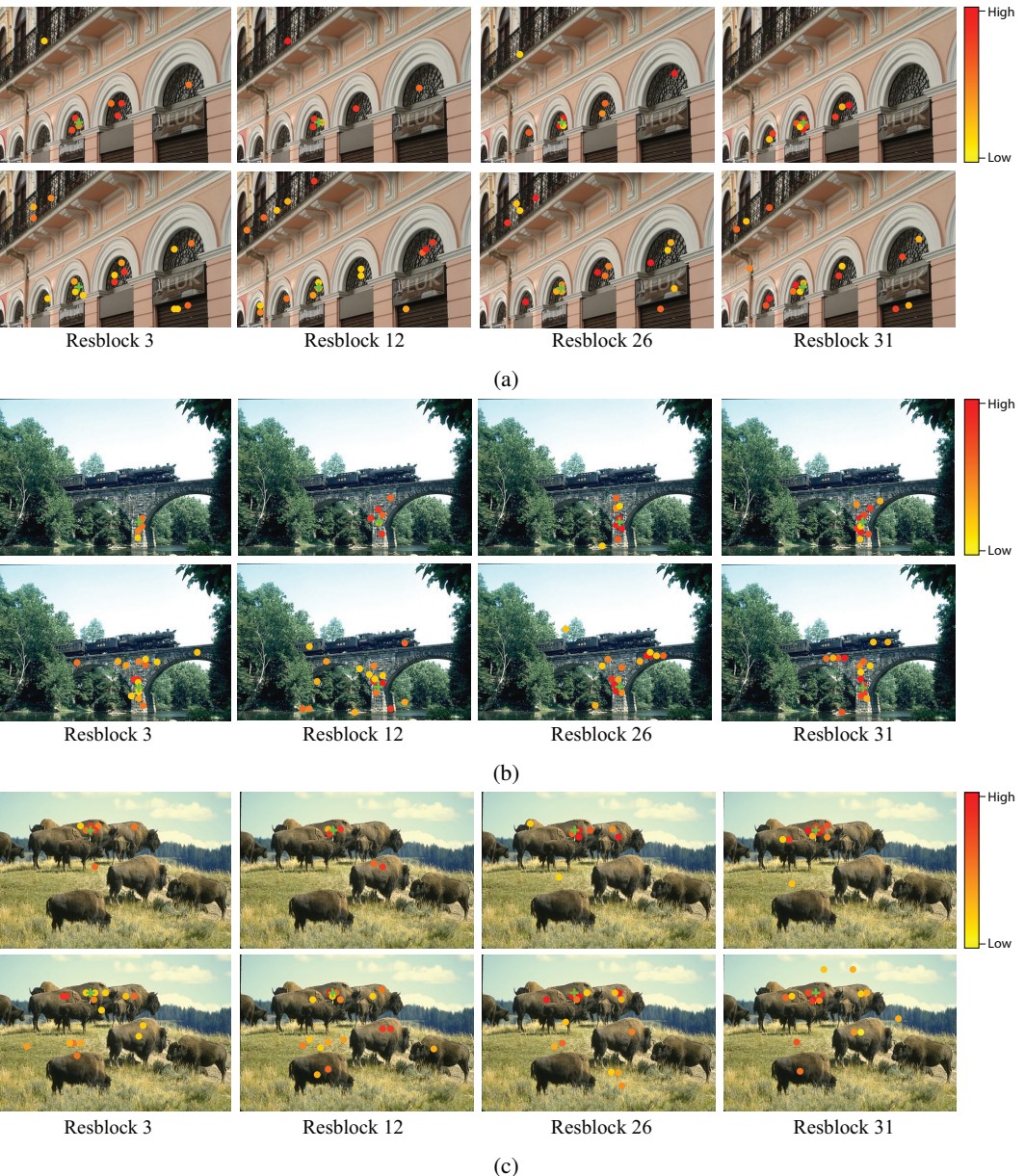

Figure 4: Visualization of selected keys by ACLA

### A.6 VISUAL RESULTS

We present some visual results of our proposed methods for 4× SR with BI degradation model in Figure 5. The visual results of CLA and ACLA are from our experiments with EDSR backbone.

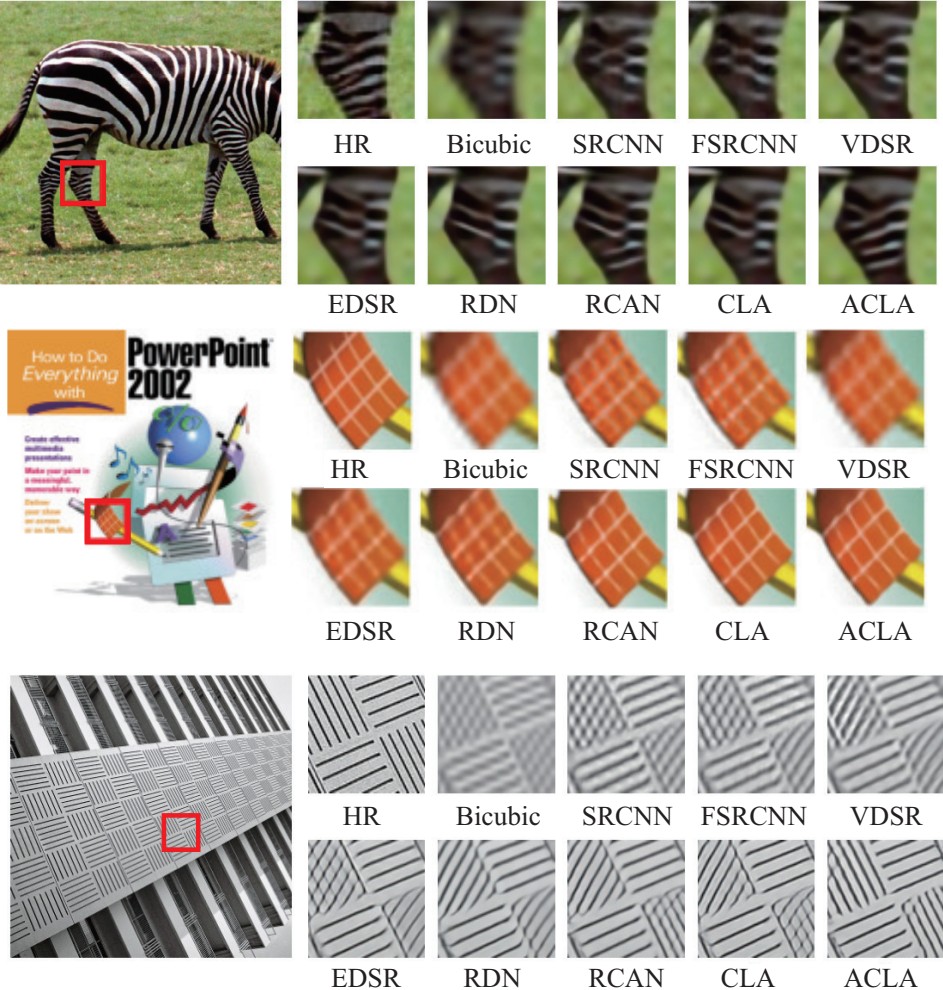

Figure 5: Visual comparison for 4× SR with BI degradation model.

