# OpenReview forum: "Adaptive Cross-Layer Attention for Image Restoration"
_ICLR.cc/2022/Conference — ICLR 2022 Submitted_

### Official Review · Reviewer_Xgo2 · 2021-10-21

**Correctness:** 3
**Technical Novelty And Significance:** 3
**Empirical Novelty And Significance:** 3
**Recommendation:** 5
**Confidence:** 5

**Main Review:**

Extensive experiments are conducted to validate the proposed method both in quality and quantity.

(1)	The effective of layer attention is a commonly used method in high-level and low-level computer vision task. There are various layer attention module was proposed in the past three years. So, the main contribution of the paper is lack of novelty. More importantly, the findings of the paper has been general knowledge in computer vision community. The authors may need to focus on more important and valuable problems in image restoration field.

(2)	The claim of “However, LAM neglects the difference of spatial positions of features.” is not strictly correct. The authors have used the CSAM to exploit the spatial positions of features in their paper. The authors may also need to compare the proposed to LAM.

(3)	The claim of “both IPT and SwinIR requires large-scale datasets for good performance.” is not exactly correct. SwinIR only use a small training set to train their network.

(4)	More importantly, in the experiments, there are only the proposed CLA and ACLA results ,which can’t present the comparative experiment with other attention mechanisms. The author may need explain the difference between the proposed CLA and other attention mechanisms in computer vision field.

(5)	The equation (13) should be further expained clearly.

(6)	In the experiments ,there are lack of params such as single image super-resolution、image denoising and image compression artifacts reduction. Also,the RCAN+CLA may neglect by the authors.



**Summary Of The Paper:**

The paper proposes a Cross-Layer Attention(CLA) module in order to capture the correlations. Between features among different layers. Besides, an Adaptive Cross-Layer Attention (ACLA) is proposed to reduce the computational cost of the Cross-Layer Attention(CLA) module. Lastly,a neural architecture search method is used to find the insert positions of ACLA modules to further improve performance.

**Summary Of The Review:**

Please see the paper weakness.

---

> ### Author Response · Authors · 2021-11-23
> **Response to Reviewer Xgo2 Part 1**
>
> (1) Problem of vanilla non-local attention and the novelty of this paper
>
> 1 Problem of vanilla non-local attention (NL)
> With non-local (NL) attention, only correlation among features at the same layer is calculated. As CNN has a hierarchical structure, features at different layers encode variant information at different scales. Besides, works like Feature Pyramid Networks[4] have shown the benefit of exploring the information from features at different layers of CNN.
>
> 2 The novelty of CLA and ACL In order to address the first problem of NL caused by only referring to keys within the same layer in most attention modules, we propose a novel attention module termed Cross-Layer Attention (CLA), which searches for keys across different layers for each query feature. With the help of the deformation mechanism, CLA only attends to a small set of keys at different layers for each query feature. Besides, we propose an improved CLA termed Adaptive Cross-Layer Attention (ACLA) which selects an adaptive number of keys at each layer for each query and searches for the optimal insert positions of ACLA modules.
> The ability to extract variant information at different layers by ACLA is proved by visualization results as in Figure 2 and Figure 4 in our paper. The visualization results show that ACLA adaptively selects semantically similar keys for the query feature, and its vanilla counterpart CLNL lacks such capability. For instance, in Figure 2, the query is from the ear of the elephant on the right side. With ACLA, 60% of the selected keys across are also from the ear of the same elephant. Besides, among the keys selected outside the ear of the same elephant, 5 out of 11 are from the ear of the elephant on the left, which have similar textures as the ear of the elephant on the right. While with CLNL, only 39% of the selected keys are from the ear of the elephant on the right. Similar observations can also be found in Figure 4. In Figure 4 (a), we pick a query point from the frame structure at the top of a gate. With ACLA, 90% of the keys selected are distributed on the frame structures at the top of the gates. While with CLNL, positions from the gates and the frame structure at the balcony are also given high attention weights. Only 61% of the selected keys are distributed on the frame structures at the top of gates, which may limit the power of attention modules. Similar observations can also be found in Figure 4 (b) and Figure 4 (c). In Figure 4 (b), the query is from the bridge in the middle of the image. All the keys selected by ACLA are also from the bridge. In Figure 4 (c), the query is from the back of a yak. Most of the keys selected by ACLA are also located on the body of yaks. While as shown in the second row, CLNL even assigns large attention weights to positions from the grass and the background. Such observations strongly demonstrate the power of ACLA in searching for informative keys across different layers.
>
> **To the best of our knowledge, our model is the first model in the computer vision community that takes account for cross-layer features and adaptive key selection at different layers for each query in attention-based neural networks, with superior performance for a broad range of low-level vision tasks including single-image super-resolution, image denoising, image compression artifacts reduction, and image demosaicing.**
>
> Most existing works in computer vision directly use features at different intermediate layers of CNN with a feature pyramid structure for the task-specific objectives [1,2,3,4]. Though attention modules have been used to process the features in the feature pyramid [2, 4], they usually simply concatenate the output of the attention modules and the output of the CNN backbone.
>
> (2) The statement “LAM neglects the difference of spatial positions of features” is true. However, we agree with your point. This statement has been revised to “While LAM neglects the difference of spatial positions of features, the HAN model indeed considers spatial positions of features in Channel-Spatial Attention Module (CSAM).” The comparison with HAN, which comprises LAM and CSAM, was presented in Table 1.
>
> (3) We have corrected the statement in section 2 of our paper. The difference between transformer-based methods and CNN-based methods is stated more clearly.

---

> > ### Author Response · Authors · 2021-11-23
> > **Response to Reviewer Xgo2 Part 2**
> >
> > (4)  In our paper, we have compared CLA and ACLA with non-local attention in section 4.5. Here, we further compare our proposed ACLA against other forms of attention modules that are widely used in the CV community, including Squeeze-and-Excitation (SE) [5] attention and Multi-Head Attention (MHA) [6]. SE aim at reweighting the channel-wise responses by using soft self-attention to model interdependencies between the channels of the convolutional features. MHA is actually a variant of self-attention from the NLP domain. Specifically, MHA can be regarded as a special kind of non-local attention that takes account of the relative position information. For the experiments with SE and MHA, we insert four SE blocks and MHA blocks evenly in the EDSR backbone, respectively. The comparison is performed on 2x single-image super-resolution following the settings in section 4.2. The results are displayed in the table below. Although MHA and SE  bring improvements over the EDSR baseline. The best results are achieved by our proposed ACLA. Besides, we also perform an experiment that combines ACLA with SE, where SE blocks are inserted after each ACLA module.
> >
> >
> > |     Methods      | Params(M) | FLOPs(G) | Set 5 | Set 14 | B 100 | Urban 100 | Manga 100 |
> > | -----------| -----|----|----|----|---|-------|-------|
> > |       EDSR       |   40.73   |  93.97   | 38.11 | 33.92  | 32.32 |   32.93   |   39.10   |
> > |    EDSR + MHA    |   42.17   |  100.21  | 38.23 | 34.01  | 32.39 |   33.07   |   39.29   |
> > |    EDSR + SE     |   41.79   |  96.14   | 38.19 | 34.03  | 32.36 |   33.06   |   39.22   |
> > |   EDSR + ACLA    |   42.29   |  96.97   | 38.31 | 34.10  | 32.43 |   33.35   |   39.42   |
> > | EDSR + ACLA + SE |   43.47   |  99.32   | 38.33 | 34.09  | 32.44 |   33.38   |   39.46   |
> >
> >
> >
> > (5) We have explained Equation (13) in section 3.2 of the revised paper.
> >
> > (6) We have added the number of parameters and the experiment for RCAN+CLA in the revised paper.
> >
> > References
> >
> > [1] Lin, Tsung-Yi, et al. "Feature pyramid networks for object detection." Proceedings of the IEEE conference on computer vision and pattern recognition. 2017.
> >
> > [2] Li, Hanchao, et al. "Pyramid attention network for semantic segmentation." arXiv preprint arXiv:1805.10180 (2018).
> >
> > [3] Kirillov, Alexander, et al. "Panoptic feature pyramid networks." Proceedings of the IEEE/CVF Conference on Computer Vision and Pattern Recognition. 2019.
> >
> > [4] Mei, Yiqun, et al. "Pyramid attention networks for image restoration." arXiv preprint arXiv:2004.13824 (2020).
> >
> > [5] Hu, Jie, Li Shen, and Gang Sun. "Squeeze-and-excitation networks." Proceedings of the IEEE conference on computer vision and pattern recognition. 2018.
> >
> > [6] Bello, Irwan, et al. "Attention augmented convolutional networks." Proceedings of the IEEE/CVF international conference on computer vision. 2019.

---

### Official Review · Reviewer_HQYo · 2021-10-27

**Correctness:** 4
**Technical Novelty And Significance:** 2
**Empirical Novelty And Significance:** 2
**Recommendation:** 5
**Confidence:** 2

**Main Review:**

Strength

- The proposed method seems quite new, and properly designed.

- The proposed method is applied to various image restoration tasks to show the generality power of the method.

Weakness

- Isn't just nonlocal attention good enough? What is the exact problem of nonlocal attention?  An experimental analysis is needed to determine the specific benefits of using cross-layer nonlocal attention. In particular, what are the advantages of using it for image restoration tasks?

- It would be better to mention differences from LAM more specifically.

- In Table1, why is there no CLA in RCAN?

- In Table1, compared to RDN or HAN, the performance is not very good. How different is the inference time? If the speed is not very fast, I think that the meaning of the proposed method is not great.

- In Table2, and Table3, similar to SR, there is not much difference in performance compared to the existing method, and the baseline model is already quite good. It is interpreted that the effect of the proposed method is not large.

- Why is the order of HAN and RDN in x4 different from x2 and x3?

- In Table 5, the difference in performance according to L is not very large, but on the contrary, FLOP and Pram increase significantly. It seems that the effect of the CLA module is not so great.

- In other ablation studies, various experiments are performed, but there is little difference in performance between each, making meaningful analysis difficult.


**Summary Of The Paper:**

This paper proposes a cross-layer attention module for the image restoration tasks. Unlike previous conventional non-local attention approaches that find correlated keys within the same layer, the proposed method selects keys across the layers. In order to prevent expensive computational costs, the authors propose adaptive cross-layer attention modules. The proposed method is validated on various image restoration tasks including SR, denoising, demosaicing, and compression.


**Summary Of The Review:**

- This paper is technically sound and quite new. However, the performance difference according to the proposed method is not significant in most experiments.

- In other words, it is difficult to say that the value of the proposed method is that high. Even technically, I think that difference with the existing methods is not significant. (Of course, the proposed method have different points compared to existing methods)

---

> ### Author Response · Authors · 2021-11-23
> **Response to Reviewer HQYo Part 1**
>
> (1) Problem of vanilla non-local attention and the novelty of this paper
>
> 1 Problem of vanilla non-local attention (NL)
> With non-local (NL) attention, only correlation among features at the same layer is calculated. As CNN has a hierarchical structure, features at different layers encode variant information at different scales. Besides, works like Feature Pyramid Networks[3] have shown the benefit of exploring the information from features at different layers of CNN.
>
> 2 The novelty of CLA and ACLIn order to address the first problem of NL caused by only referring to keys within the same layer in most attention modules, we propose a novel attention module termed Cross-Layer Attention (CLA), which searches for keys across different layers for each query feature. With the help of the deformation mechanism, CLA only attends to a small set of keys at different layers for each query feature. Besides, we propose an improved CLA termed Adaptive Cross-Layer Attention (ACLA) which selects an adaptive number of keys at each layer for each query and searches for the optimal insert positions of ACLA modules.
>
> The ability to extract variant information at different layers by ACLA is proved by visualization results as in Figure 2 and Figure 4 in our paper. The visualization results show that ACLA adaptively selects semantically similar keys for the query feature, and its vanilla counterpart CLNL (Cross-Layer Non-Local) lacks such capability. For instance, in Figure 2, the query is from the ear of the elephant on the right side. With ACLA, 60% of the selected keys across are also from the ear of the same elephant. Besides, among the keys selected outside the ear of the same elephant, 5 out of 11 are from the ear of the elephant on the left, which have similar textures as the ear of the elephant on the right. While with CLNL, only 39% of the selected keys are from the ear of the elephant on the right. Similar observations can also be found in Figure 4. In Figure 4 (a), we pick a query point from the frame structure at the top of a gate. With ACLA, 90% of the keys selected are distributed on the frame structures at the top of the gates. While with CLNL, positions from the gates and the frame structure at the balcony are also given high attention weights. Only 61% of the selected keys are distributed on the frame structures at the top of gates, which may limit the power of attention modules. Similar observations can also be found in Figure 4 (b) and Figure 4 (c). In Figure 4 (b), the query is from the bridge in the middle of the image. All the keys selected by ACLA are also from the bridge. In Figure 4 (c), the query is from the back of a yak. Most of the keys selected by ACLA are also located on the body of yaks. While as shown in the second row, CLNL even assigns large attention weights to positions from the grass and the background. Such observations evidence the power of ACLA in searching for informative keys across different layers.
>
> (2) Layer attention module (LAM) is proposed in holistic attention network (HAN) [1] for single image super-resolution. LAM is designed to explore the correlation among features at different layers. Specifically, the features from $N$ layers are reshaped into $N\times HWC$, where $H$, $W$, and $C$ are the height, width, and channel number of the feature map. A $N \times N$ correlation matrix is computed among the features of $N$ layers. Thus, the correlation weights are the same for different spatial positions between two different layers. In contrast, ACLA selects adaptive keys at different spatial positions across different layers by a dynamic gating module based on Gumbel-Softmax.
>
> (3) We have added the experiments for RCAN + CLA for single image super-resolution. The results have been added to Table 5 of the revised paper.
>
> (4) Here we compare the inference time between our proposed ACLA and previous state-of-the-arts methods. The running time is the average of 1000 runs on inputs of size $48\times48$. The running time is evaluated on a single 16G Tesla V100. We compare our proposed methods with HAN [1] and SAN [4], which are also attention-based methods for single image super-resolution. As shown in the table, our method EDSR+ACLA achieves even better performance than HAN with much less inference time.
>
> |                    | RCAN+ACLA | EDSR+ACLA | RCAN  | HAN [1] | SAN [4] |
> |--------------|----|-----|--|----|---|
> |    Set 5 (PSNR)    |   38.30   |   38.31   | 38.27 |  38.27  |  38.31  |
> | Inference time(ms) |   36.9    |   19.8    | 32.7  |  38.9   |  61.2   |

---

> > ### Author Response · Authors · 2021-11-23
> > **Response to Reviewer HQYo Part 2**
> >
> > (5) As image restoration is regarded as an ill-posed problem, improving the performance of CNN backbones for image restoration has always been a challenging task. Recently, attention methods have been widely used to improve performance. Here, we compare our ACLA with HAN [1] and SAN [4], which are also attention-based methods for single image super-resolution. Both HAN and SAN are based on the previous SOTA method RCAN. In the table below, we compared the improvements of ACLA, HAN, and SAN. The improvements over RCAN are listed in the parentheses after the PSNR results.   For our ALCA, the percentage comparisons of improvement are also calculated. For example, the improvement of ACLA over RCAN on B100 is 0.04 dB, which is 400% of the improvement of SAN over RCAN. Besides, we have also calculated the improvement by ACLA on the EDSR backbone. As shown in the table, compared to HAN and SAN which are competitive baselines representing the recent progress of this literature, our method makes significant improvements in this literature.
> >
> >
> > |  Methods  |         Set 5          |         Set 14         |          B100           |        Urban100         |        Manga109         |
> > |------|-------------|----------|-----------|---------|-----------|
> > |   RCAN    |         38.27          |         34.12          |          32.41          |          33.34          |          39.44          |
> > |    HAN    |      38.27 (0.00)      |      34.16 (0.04)      |       32.41(0.00)       |      33.35 (0.01)       |      39.46 (0.02)       |
> > |    SAN    |      38.31 (0.04)      |     34.07 (-0.05)      |      32.42 (0.01)       |      33.10 (-0.24)      |      39.32 (-0.12)      |
> > | RCAN+ACLA | 38.30 (0.03 / **75%**) | 34.15 (0.03 / **75%**) | 32.45 (0.04 / **400%**) | 33.39 (0.05 / **500%**) | 39.48 (0.04 / **200%**) |
> > |   EDSR    |         38.11          |         33.92          |          32.32          |          32.93          |          39.10          |
> > | EDSR+ACLA |      38.31 (0.20)      |      34.10 (0.18)      |      32.43 (0.11)       |      33.35 (0.42)       |      39.42 (0.32)       |
> >
> >
> > Additionally, as shown in part (4) of our response to Review Xgo2, we have conducted more comparative results between our method and other attention mechanisms.
> >
> > (6) We have fixed that and the order of results for x2, x3, and x4 is now consistent in the revised version.
> >
> > (7) In previous works on non-local attention [2], similar observations were also found. Specifically, sparsely inserting non-local attention modules in CNNs can already yield good results. Further increasing the number of inserted non-local attention only brings marginal improvement while increasing the inference cost. For example, in Non-local Neural Networks [2] inserting 5 non-local blocks in ResNet yields a similar performance as inserting 10 non-local blocks. In our experiments with CLA, similar results are concluded with our ablation study on the number of inserted CLA modules. Thus, we only use EDSR models with 4 CLA modules inserted for our experiments. Besides, this also motivates us to search for the insert positions in ACLA with a constraint on inference cost.

---

> > > ### Author Response · Authors · 2021-11-23
> > > **Response to Reviewer HQYo Part 3**
> > >
> > > (8) The significance of our results in the ablation study is explained here in a similar manner to part (5) of this response.
> > >
> > > For the ablation study in Table 4, we aim to compare our proposed CLA and ACLA with NL. In the table below, we compare the improvements of CLNL, CLA, and ACLA over the baseline with NL. The percentage comparisons of improvement are also calculated and listed in the parentheses after the PSNR results.  For example, the improvement of ACLA over NL on Set 5 is 0.16 dB, which is 400% of the improvement of SAN over its baseline RCAN. As shown in the table, compared to HAN and SAN, our proposed CLA and ACLA make significant improvements over the baseline attention module NL.
> > >
> > > | Methods |          Set 5           |         Set 14          |          B100           |         Urban100         |         Manga109         |
> > > |----|------------|----------|------------|---------------|----------|
> > > |  RCAN   |          38.27           |          34.12          |          32.41          |          33.34           |          39.44           |
> > > |   HAN   |       38.27 (0.00)       |      34.16 (0.04)       |       32.41(0.00)       |       33.35 (0.01)       |       39.46 (0.02)       |
> > > |   SAN   |       38.31 (0.04)       |      34.07 (-0.05)      |      32.42 (0.01)       |      33.10 (-0.24)       |      39.32 (-0.12)       |
> > > |   NL    |          38.15           |          34.01          |          32.35          |          33.07           |          39.19           |
> > > |  CLNL   | 38.14 (-0.01 / **-25%**) | 34.04 (0.03 / **75%**)  | 32.38 (0.03 / **300%**) | 33.13 (0.06 / **600%**)  |  39.20 (0.01 / **50%**)  |
> > > |   CLA   | 38.24 (0.09 / **225%**)  | 34.08 (0.01 / **175%**) | 32.41 (0.06 / **600%**) | 33.28 (0.21 / **2100%**) | 39.23 (0.04 / **200%**)  |
> > > |  ACLA   | 38.31 (0.16 / **400%**)  | 34.10 (0.09 / **225%**) | 32.43 (0.08 / **800%**) | 33.35 (0.28 / **2800%**) | 39.42 (0.23 / **1150%**) |
> > >
> > >
> > > For the ablation study in Table 6, we aim to study the effectiveness of two adaptive designs applied in ACLA. In the table below, we compare the improvements of CLA-I, CLA-K, and ACLA over the baseline with CLA. The percentage comparisons of improvement are also calculated and listed in the parentheses after the PSNR results.  For example, the improvement of CLA-K over CLA on Manga109 is 0.14 dB, which is 800% of the improvement of SAN over its baseline RCAN. As shown in the table, compared to the improvements by HAN and SAN, both adaptive designs applied in ACLA make significant improvements over CLA.
> > >
> > >
> > > | Methods |          Set 5          |         Set 14         |          B100           |        Urban100         |        Manga109         |
> > > |----|------------|----------|------------|---------------|----------|
> > > |  RCAN   |          38.27          |         34.12          |          32.41          |          33.34          |          39.44          |
> > > |   HAN   |      38.27 (0.00)       |      34.16 (0.04)      |       32.41(0.00)       |      33.35 (0.01)       |      39.46 (0.02)       |
> > > |   SAN   |      38.31 (0.04)       |     34.07 (-0.05)      |      32.42 (0.01)       |      33.10 (-0.24)      |      39.32 (-0.12)      |
> > > |   CLA   |          38.24          |         34.08          |          32.41          |          33.28          |          39.23          |
> > > |  CLA-I  | 38.27 (0.03 / **75%**)  | 34.10 (0.02 / **50%**) | 32.42 (0.01 / **100%**) | 33.31 (0.03 / **300%**) | 39.35 (0.12 / **600%**) |
> > > |  CLA-K  | 38.28 (0.04 / **100%**) | 34.09 (0.01 / **25%**) | 32.43 (0.02 / **200%**) | 33.31 (0.03 / **300%**) | 39.37 (0.14 / **800%**) |
> > > |  ACLA   | 38.31 (0.07 / **175%**) | 34.10 (0.02 / **50%**) | 32.43 (0.02 / **200%**) | 33.35 (0.07 / **700%**) | 39.42 (0.19 / **950%**) |
> > >
> > >
> > > For ablation study on the number of inserted CLA modules from Table 4. The analysis of the results has been addressed in response (7). For ablation study on the number of sampled keys, we aim to verify that a small number of sampled keys can be sufficient for our proposed CLA and ACLA. The results show us that CLA with $K=8$ and ACLA with $K=16$ can already achieve comparable performance to those with larger $K$.
> > >
> > > References:
> > >
> > > [1] Niu, Ben, et al. "Single image super-resolution via a holistic attention network." European Conference on Computer Vision. Springer, Cham, 2020.
> > >
> > > [2] Wang, Xiaolong, et al. "Non-local neural networks." Proceedings of the IEEE conference on computer vision and pattern recognition. 2018.
> > >
> > > [3] Lin, Tsung-Yi, et al. "Feature pyramid networks for object detection." Proceedings of the IEEE conference on computer vision and pattern recognition. 2017.
> > >
> > > [4] Dai, Tao, et al. "Second-order attention network for single image super-resolution." Proceedings of the IEEE/CVF Conference on Computer Vision and Pattern Recognition. 2019.

---

### Official Review · Reviewer_6N93 · 2021-10-31

**Correctness:** 3
**Technical Novelty And Significance:** 3
**Empirical Novelty And Significance:** 2
**Recommendation:** 8
**Confidence:** 4

**Main Review:**

Attention is a powerful mechanism and has found utility in numerous tasks in both high and low level computer vision as effectively argued in the paper.  Therefore paper is addressing an interesting topic relevant to the ICLR community.

Note however, the idea of exploiting correlations across layers in convolutional networks isn’t new, for example, the papers

•	Chang et al., “EPSNet: Efficient Panoptic Segmentation Network with Cross-layer Attention Fusion,” ACCV 2020 also applies non-local attention at different layers, in a hierarchical network architecture used for image segmentation.

•	Mei et al., “Image Super-Resolution with Cross-Scale Non-Local Attention and Exhaustive Self-Exemplars Mining,” CVPR 2020 from the authors of the PANet (Mei et al., 2020) referenced and compared to in the paper; has similar motivations for image restoration tasks.

However, the approach taken in this paper appears to be novel through its use of deformable convolution, gating, and neural architecture search to reduce the computational complexity.

The paper is clearly described.  It would be helpful if the paper commented on the availability of source code in order for others to replicate the results.  The approach is fairly generic so the method may find applicability in many tasks beyond image restoration.

Experimental results cover a wide range of image restoration tasks to prove the effectiveness and efficiency of the proposed module.  Although the method often produces the best results, the incremental improvement is arguably small; for example in Table 4 the PSNR increases by 0.2 dB compared to EDSR.  However, the visual results are compelling particularly in the appendix.  It seems somewhat surprising to this reviewer that the PSNR improvement is so small yet visually the results appear to be more pronounced (for example, in Figure 5).

Strengths:
•	While cross-layer attention isn’t a new idea, the approach taken in this paper has novelty and provides a fairly generic approach that could be used with different backbones.
•	The method carefully addresses the issue of computational complexity to produce a practical working solution.

•	Comprehensive experiments including quantitate analysis and ablation study shows how the proposed algorithm performs efficiently.

•	The paper is clearly written and includes useful figures, e.g. Figure 2.

Weaknesses.
•	The primary disadvantage is the small incremental benefit for some of the steps of the method.  For example, the ablation study in Table 6 shows only a 0.07 dB improvement on Set5 for ACLA compared to CLA.  Given that ACLA requires neural architecture search, dynamic key selection and more it may be a lot of work for limited benefit to squeeze out less than 0.1 dB improvement.

Small corrections and suggestions:

•	The first sentence in the introduction describes degradation as “irreversible”.  But the idea of the image restoration is to restore the degradation.  If the image is restored, isn’t the degradation reversed?  Perhaps this qualifying statement could be removed or clarified.

•	Page 1 argues that most CNN-based image restoration methods tend to produce smooth results, potentially due to only referring to keys in the same layer in their attention modules.  This seems highly speculative – it would be better to show this is the case or possibly remove this.  It’s well known there is a perception-distoration tradeoff in image restoration; e.g. Blau et al., “The Perception-Distortion Tradeoff,” CVPR 2018 and blurring often results in minimizing distortion using standard losses and can be addressed through generative methods that hallucinate detail at the expense of PSNR.

•	Page 4, please change “To find for keys” to “To search for keys”

•	Page 5, please change “ALDA” to “ACLA”

•	Page 5, please change “top op CLA” to “top of CLA”


**Summary Of The Paper:**

This paper proposes a novel Cross Layer Attention (CLA) module for image restoration tasks. The CLA module doesn’t look for correlated key pixels within the same layer as other algorithms do; instead query pixels attend to key pixels at previous layers of the network. To reduce computational complexity deformable convolution is used to reduce the number of sampled keys.  To further reduce computational complexity, the paper proposes a modified CLA called Adaptive Cross Layer Attention (ACLA). In ACLA, the number of collected keys for each query is dynamically selected through a gating mechanism.  Further, a neural architecture search method was used to find the proper positions to insert the ACLA modules in the backbone network.  The paper provides comprehensive experiments on image restoration tasks to validate the effectiveness and efficiency of ACLA. Conducted experiments include single image super resolution, image denoising, image demosaicing and image compression artifact reduction.  Contributions are the CLA and ACLA modules designed to efficiently perform attention across layers in a neural network.

**Summary Of The Review:**

Overall, this paper address an important problem of performing attention across different layers in a neural network and proposes a novel approach that carefully handles the computational complexity.  The method is demonstrated to work effectively in several image restoration methods.  Numerically the results are only slightly better than competitors, but it does reflect an advance.  This reviewer is somewhere between marginally above and accept for this paper.

---

> ### Author Response · Authors · 2021-11-23
> **Response to Reviewer 6N93**
>
> (1) “The primary disadvantage is the small incremental benefit for some of the steps of the method.”
> Please refer to part (5) of our response to reviewer HQYo regarding the significance of our results when compared to the most recent baselines in this literature.
>
> (2) “The first sentence in the introduction describes degradation as ‘irreversible’...”
> Thank you for the correction. We have corrected the statement in the revised paper. The image restoration process aims at recovering high-quality images from low-quality images. What we intend to state is that image restoration is usually viewed as an ill-posed problem, since there is no unique solution associated with it.
>
> (3) “Page 1 argues that most CNN-based image restoration methods tend to produce smooth results...”
> Thank you for the insightful point. In our paper, we use MSE loss for the training and evaluate our model with distortion measures, which are PSNR and SSIM. We have removed the statement in the introduction of our revised paper. The motivation behind our cross-layer attention design is that features at different layers of CNN encode variant information at different scales, as features at deeper layers have a larger receptive field. With our proposed CLA and ACLA, we aim to take account of information at different scales to augment the information used in recovering the high-quality image.
>
> (4) We have fixed the typos in the revised version of our paper.

---

### Official Review · Reviewer_g1nJ · 2021-12-11

**Correctness:** 3
**Technical Novelty And Significance:** 2
**Empirical Novelty And Significance:** 3
**Recommendation:** 6
**Confidence:** 4

**Main Review:**

Strengths:
(1)	This work presents cross-layer attention (CLA) modules and adaptive cross-layer attention (ACLA) to model non-local pixel correlations by considering feature correlations among different layers.
(2)	Experimental results show that ACLA works well for different image restoration tasks.
Weaknesses:
1.	In Tables 1, 2, & 3, the authors are suggested to add an experiment to replace CLA with the classical non-local block.
2.	It is unclear why finding the correlated key pixels from previous CNN layers is capable to enhance the image restoration performance.
3.	Compared to CLA, the ACLA dynamically selects key pixels by using a NAS. However, according to Table 1, ACLA only slightly improves CLA in terms of super-resolution accuracy. It tends to degrade the effectiveness of the NAS of ACLA. Please discuss it.


**Summary Of The Paper:**

This work presents cross-layer attention (CLA) modules to find informative keys across different CNN layers for each query feature. Furthermore, an adaptive cross-layer attention (ACLA) is also formulated to dynamically select keys from different CNN layers by using a NAS method. After that, the authors embedded the presented CLA or ACLA modules into EDSR to formulate a deep model for image restoration. Experimental results on several image restoration tasks show that the developed deep model outperforms state-of-the-art methods.

**Summary Of The Review:**

Please refer to the weaknesses of the main review.

---

### Author Response · Authors · 2021-12-03
**We greatly appreciate the reviews from all the reviewers and look forward to further comments**

Dear Reviewers and AC,

We would like to express our gratitude for your time and efforts in reviewing and handling this paper. We have responded to the concerns of all the reviewers and revised our paper accordingly. We would greatly appreciate it if Reviewer HQYo could look into our response to your last comment, which is titled "**Further response regarding the significance of our results**" where the statistical significance of the improvement of our method over previous SOTA with variance and p-value on all the four tasks is demonstrated. We also appreciate it if Reviewer Xgo2 could let us know if there are remaining issues after reading our response.

Again, thank all of you for your time!

Best Regards,

Authors of Paper3241

---

### Decision · Program_Chairs · 2022-01-20

**Decision:**

Reject

**Comment:**

The paper introduces an cross-layer attention mechanism for image restoration. To reduce the computational complexity, the framework uses deformable convolutions and an adaptive selection for reducing the number of keys, as well as a neural architecture search. The paper received three borderline reject recommendations and a clear accept. After reading the reviews, responses, and the paper in details, the area chair agrees with Reviewer 6N93 that the paper has some merit. Unfortunately, he/she also agrees with the fact that the proposed framework is quite complicated with many components for a marginal improvement (something that also Reviewer 6N93 has mentioned in the discussion between reviewers). Overall, this points towards rejection, which is the final recommendation of the area chair.

Another point that would be helpful, in case this paper is resubmitted elsewhere, is to release the code for the method, given its complexity.